# Towards Context-Aware Domain Generalization: Understanding the Benefits and Limits of Marginal Transfer Learning

**Jens Müller** *
Independent Researcher
Germany

**Lars Kühmichel** *
Computational Statistics
TU Dortmund University
Germany
`lars.kuehmichel@tu-dortmund.de`

**Martin Rohbeck**
Division of Computational Genomics and Systems Genetics
DKFZ
Germany

**Stefan T. Radev**
Department of Cognitive Science
Rensselaer Polytechnic Institute
NY, USA

**Ullrich Köthe**
Computer Vision and Learning Lab
Heidelberg University
Germany

## Abstract

In this work, we analyze the conditions under which information about the context of an input data point can improve the predictions of deep learning models in new domains. Following work in marginal transfer learning and domain generalization, we formalize the notion of context as a permutation-invariant representation of a set of data points that originate from the same domain as the input itself. We offer a theoretical analysis of the conditions under which this approach can, in principle, yield benefits, and formulate two necessary criteria that can be easily verified in practice. Additionally, we contribute insights into the kind of distribution shifts for which the marginal transfer learning approach promises robustness. Empirical analysis shows that our criteria are effective in discerning both favorable and unfavorable scenarios. Finally, we demonstrate that we can reliably detect scenarios where a model is tasked with unwarranted extrapolation in out-of-distribution (OOD) domains, identifying potential failure cases. Consequently, we showcase a method to select between the most predictive and the most robust model, circumventing the well-known trade-off between predictive performance and robustness.

## 1 Introduction

Distribution shifts are the cause of many failure cases in machine learning [1, 2] and the root of various peculiar phenomena in classical statistics, such as Simpson's paradox [3, 4]. Domain Generalization (DG) seeks models that are robust to distribution shifts by utilizing data from distinct environments during training [5, 6]. In the context of DG, *marginal transfer learning* enhances a model with context

---

*Equal Contribution

39th Conference on Neural Information Processing Systems (NeurIPS 2025) Workshop: Reliable ML From Unreliable Data.

**A) Data-Generating Process**  **B) Context-Aware Domain Generalization**

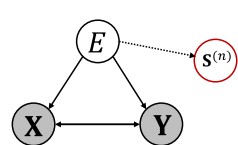 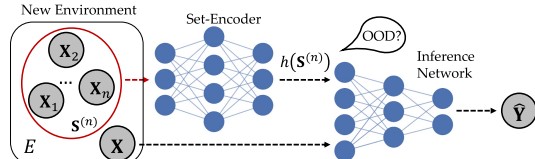

Figure 1: **Conceptual sketch of our setup and approach**. **A)** Data-generating process (DGP) that fulfills our criteria. We assume that the environment $E$ is a source node that is not caused by any system variable and that the relationship between $\mathbf{X}$ and $\mathbf{Y}$ varies with the environment. $\mathbf{S}^{(n)}$ is a set of $n$ i.i.d. inputs available in the new environment. The bidirectional arrow indicates that the causal relation between $\mathbf{X}$ and $\mathbf{Y}$ could be explained by a common cause or $\mathbf{Y}$ causing $\mathbf{X}$ (or vice versa). **B)** The context-aware model (marginal transfer learning approach) in a test environment. A set-encoder generates a permutation-invariant representation $h(\mathbf{S}^{(n)})$ of the context. An inference network processes the representation along with the target input $\mathbf{X}$ and predicts the unknown outcome of the target input. The set-representation can be combined with the input to reliably detect out-of-distribution queries and prevent failure cases in domain generalization due to model misspecification.

information to achieve better predictions [7]. The "context" of a test instance is a set of samples that stems from the same environment as the instance itself and can be embedded, for instance, by permutation-invariant neural networks [8]. In this work, we enhance the fundamental understanding of settings where marginal transfer learning in DG can reap benefits compared to baseline models.

Consider a probabilistic model $p(\mathbf{Y} \mid \mathbf{X})$ that classifies diseases $\mathbf{Y}$ from magnetic resonance (MR) images $\mathbf{X}$. Since MR images are not fully standardized, the classifier should work slightly differently for images acquired by different hardware brands. It thus makes sense to inform the classifier about the current environment $E$ (here: hardware brand) and extend it into $p(\mathbf{Y} \mid \mathbf{X}, E)$. This raises a key question: Under which circumstances will the classifier $p(\mathbf{Y} \mid \mathbf{X}, E)$ be superior to $p(\mathbf{Y} \mid \mathbf{X})$? This question is important because there might exist a function $E = f(\mathbf{X})$ allowing the classifier $p(\mathbf{Y} \mid \mathbf{X})$ to deduce $E$ from the data $\mathbf{X}$ alone. For example, $E$ might be inferred from the periphery of the given image, while $\mathbf{Y}$ depends on its central region. Then, no additional information is gained by passing $E$ explicitly, and both classifiers perform identically.

Building on previous work in marginal transfer learning [7], we aim to learn a continuous embedding of $E$ from auxiliary data using set-encoders, as depicted in Figure 1. We then establish three criteria that delineate the circumstances in which $p(\mathbf{Y} \mid \mathbf{X}, E)$ is beneficial, and subsequently prove their necessity. Notably, two of these criteria are empirically testable using standard models and are shown to be necessary conditions for the success of the approach.

When test environments are highly dissimilar to the training environments, all DG methods enter an extrapolation regime with unknown prospects of success and an increased risk of silent failures. While marginal transfer learning is not exempt from this "curse of extrapolation", we find that it comes with a natural way to reliably detect novel environments in set-representation space and delineate its competence region [9]. Accordingly, we propose a method to select between models that are specialized in the ID setting versus models that are robust to OOD scenarios on the fly. Thus, we can overcome the notorious trade-off between ID predictive performance and robustness to distribution shifts [10–12]. In summary, our contributions are:

- We formalize the necessary and empirically verifiable conditions under which the marginal transfer learning can improve on standard approaches;

- We show empirically that we can identify cases where context-aware models offer no advantages or when dangerous extrapolation is necessary;

- We show how the detection of novel environments allows for model selection, overcoming the trade-off between predictive performance and robustness.

## 2 Method

### 2.1 Notation

We denote inputs $\mathbf{X} \in \mathcal{X}$ and outputs as $\mathbf{Y} \in \mathcal{Y}$, without any strict requirements on the input and output spaces $\mathcal{X}$ and $\mathcal{Y}$, respectively. We treat the (unknown) domain label $E$ as a random variable and denote with $\mathbf{S}^{(n)}$ a set of $n$ further i.i.d. samples from a given domain, whose label $E$ is only known during training time.

### 2.2 Context-Aware Models

A context-aware model consists of two key components (also illustrated in Figure 1): (i) a permutation-invariant network $h_{\boldsymbol{\psi}}$ ("set-encoder") with parameters $\psi$ that maps a set-input $\mathbf{S}^{(n)}$ to a summary vector $h_{\boldsymbol{\psi}}(\mathbf{S}^{(n)})$, and (ii) an inference network $f_{\boldsymbol{\phi}}$ with parameters $\phi$ that maps both the input $\mathbf{X}$ and the summary vector $h_{\boldsymbol{\psi}}(\mathbf{S}^{(n)})$ to a final prediction. The complete model is denoted as $f_{\boldsymbol{\theta}}(\mathbf{X}, \mathbf{S}^{(n)}) = f_{\boldsymbol{\phi}}(\mathbf{X}, h_{\boldsymbol{\psi}}(\mathbf{S}^{(n)}))$ with parameters $\boldsymbol{\theta} = (\psi, \phi)$ for short. For a given supervised learning task, we consider the optimization problem

$$\widehat{\boldsymbol{\theta}} = \arg\min_{\boldsymbol{\theta}} \mathbb{E}_{p(\mathbf{X}, \mathbf{Y}, E)} \left[ c(f_{\boldsymbol{\theta}}(\mathbf{X}, \mathbf{S}^{(n)}), \mathbf{Y}) \right], \tag{1}$$

where $c$ is a task-specific loss function (e.g., cross-entropy for classification or mean squared error for regression). Algorithm 1 details the optimization of Equation 1.

### 2.3 Criteria for Improvement

In the following, we establish criteria under which context information allows to exploit the distribution shifts between environments and yield improved predictions.

In total, we propose three criteria that are necessary to achieve incremental improvement. In Theorem 2.1, we show how these criteria are related to each other. In the formulations below, $I(\mathbf{X}; \mathbf{Y})$ denotes the *mutual information* between random vectors $\mathbf{X}$ and $\mathbf{Y}$ and $I(\mathbf{X}; \mathbf{Y} \mid \mathbf{Z})$ denotes the conditional mutual information given a third random vector $\mathbf{Z}$. The symbol $\perp$ (resp. $\not\perp$) between two random vectors $\mathbf{X}$ and $\mathbf{Y}$ is used to express that the random vectors are independent (resp. dependent) or conditionally independent (resp. dependent) given a third random vector $\mathbf{Z}$.

First, we require that given an input $\mathbf{X}$, a further set of i.i.d. inputs $\mathbf{S}^{(n)}$ from the same environment provides *incremental information* about $\mathbf{Y}$. This is exactly what we need to achieve improved predictive performance, and we can formally define it as our first criterion:

**Criterion 2.1.** $\mathbf{S}^{(n)} \not\perp \mathbf{Y} \mid \mathbf{X}$ *or* $I(\mathbf{S}^{(n)}; \mathbf{Y} \mid \mathbf{X}) > 0$.

The second criterion requires that, given a target input $\mathbf{X}$, a set of further i.i.d. inputs $\mathbf{S}^{(n)}$ from the same environment provides *additional information* about the origin environment of $\mathbf{X}$.

**Criterion 2.2.** $E \not\perp \mathbf{S}^{(n)} \mid \mathbf{X}$ *or* $I(E; \mathbf{S}^{(n)} \mid \mathbf{X}) > 0$.

In Figure 2, an instance $\mathbf{X}$ cannot be assigned with complete certainty to an environment. Consequentially, further data provides additional information about the environment. In general, the more data we consider, the better we can predict the originating environment. Crucially, this criterion is *not satisfied* if we can recover the origin environment from the singleton input $\mathbf{X}$ alone.

The third criterion requires that the singleton input $\mathbf{X}$ carries information about $\mathbf{Y}$ if we also consider the origin environment $E$ of $\mathbf{X}$.

**Criterion 2.3.** $\mathbf{Y} \not\perp E \mid \mathbf{X}$ *or* $I(\mathbf{Y}; E \mid \mathbf{X}) > 0$.

This criterion can serve as a sanity check in case we have an oracle that can identify the origin environment of the data with perfect accuracy. In what follows, we show that Criterion 2.2 and Criterion 2.3 are necessary conditions for Criterion 2.1. We furthermore prove that if we can extract the environment label fully from $\mathbf{S}^{(n)}$, then Criterion 2.2 and Criterion 2.3 are sufficient conditions for Criterion 2.1.

**Theorem 2.1.** *The following statements hold:*

(a) If $E \perp \mathbf{S}^{(n)} \mid \mathbf{X}$, it follows that $\mathbf{Y} \perp \mathbf{S}^{(n)} \mid \mathbf{X}$. This is equivalent to the implication that if Criterion 2.2 is unattainable, then Criterion 2.1 is also not satisfied.

(b) If $E \perp \mathbf{Y} \mid \mathbf{X}$, we achieve $\mathbf{Y} \perp \mathbf{S}^{(n)} \mid \mathbf{X}$. This statement corresponds to: Criterion 2.3 is a necessary condition for Criterion 2.1.

(c) Assume that there exists a deterministic function $g$ with $g(\mathbf{S}^{(n)}) = E$, then $\mathbf{Y} \not\perp E \mid \mathbf{X}$ implies $\mathbf{Y} \not\perp \mathbf{S}^{(n)} \mid \mathbf{X}$. This conveys that if we could perfectly infer $E$ from $\mathbf{S}^{(n)}$, then Criterion 2.3 implies Criterion 2.1.

In our experiments, we observe that a function $g(\mathbf{S}^{(n)}) = E$ can already be found for small $n$ (see for instance Figure 2). In this case, we obtain $I(\mathbf{S}^{(n)}; \mathbf{Y} \mid \mathbf{X}) = I(E; \mathbf{Y} \mid \mathbf{X})$ and Criterion 2.2 and Criterion 2.3 are sufficient to obtain Criterion 2.1. Unfortunately, we cannot conclude that $Y \not\perp \mathbf{S}^{(n)} \mid \mathbf{X}$ follows from Criterion 2.2 and Criterion 2.3 in general. A counterexample where Criterion 2.2 and Criterion 2.3 hold, but Criterion 2.1 is violated, is provided in Appendix C.2. We furthermore provide the proof of the theorem, an illustration for the theorem as well as a generalization of *(c)* in Appendix C.

It is worth noting that model misspecification adds another layer of uncertainty when verifying the criteria. In cases where determining the correct mutual information is not feasible (for instance, when $p(\mathbf{Y} \mid \mathbf{X})$, $p(\mathbf{Y} \mid \mathbf{X}, \mathbf{S}^{(n)})$, or $p(\mathbf{Y} \mid \mathbf{X}, E)$ cannot be learned adequately), two primary issues may emerge. Firstly, the effective utilization of the set-input $\mathbf{S}^{(n)}$ (or E) may be hindered due to either the restricted expressive power of the model class or a scarcity of training data. As a result, the context-aware model might not improve on a baseline model that only utilizes $\mathbf{X}$. Consequentially the criteria could seem unattainable while they actually are. Secondly, after training, we might observe an apparent advantage of the approximation of $p(\mathbf{Y} \mid \mathbf{X}, \mathbf{S}^{(n)})$ or $p(\mathbf{Y} \mid \mathbf{X}, E)$ over the approximation of $p(\mathbf{Y} \mid \mathbf{X})$, despite the true model not conferring any advantage. In this scenario, the criteria may appear to be satisfied, whereas in reality they are not. An example of this case can be easily constructed by considering a non-linear $p(\mathbf{Y} \mid \mathbf{X})$ and a linear function class.

## 2.4 Source Component Shift

Using our approach, we can characterize the kind of distribution shift that allows our criteria to be satisfied. *Source component shift* refers to the scenario where the data comes from a number of sources (or environments) each with different characteristics [13]. The source component shift can be described by the graphical model in Figure 1, where the environment directly affects both the input $\mathbf{X}$ and the outcome $\mathbf{Y}$. Problems that conform to the graph in Figure 1 have two important implications. First, the input distribution changes whenever the environment changes. Second, the relationship between inputs and outcomes varies with the environment (corresponding to Criterion 2.1). For more details on this kind of distribution shift, we refer the reader to [13]. It is also worth noting that the graph in Figure 1 corresponds to Simpson's paradox [3, 4], which supplies a proof-of-concept for our approach (see **Experiment 1**). An important point to highlight is that the frequently encountered covariate shift where only $P(\mathbf{X})$ in $P(\mathbf{X}, \mathbf{Y}) = P(\mathbf{Y} \mid \mathbf{X})P(\mathbf{X})$ varies between environments [13], does not conform to the conditions specified in Criterion 2.3. Hence, context-aware models do not provide advantages when compared to standard models under covariate shift.

## 2.5 Detection of Novel Environments

During test time, data could either originate from an environment that corresponds to one of the training environments (but its origins are unknown) or from a previously unseen environment. In the following, we explain how we aim to detect the second case that might result in potential failure cases due to fundamental challenges in extrapolation. Following [9], we can define a score $s(h_\psi(\mathbf{S}^{(n)}))$ on the summary vector $h_\psi(\mathbf{S}^{(n)})$ implicit in our model $f_{\boldsymbol{\theta}}(\mathbf{X}, \mathbf{S}^{(n)})$ that aims to predict the target variable $\mathbf{Y}$. As a score function, we consider the distance of $h_\psi(\mathbf{S}^{(n)})$ to the $k$-nearest neighbors in the training data in the feature space of the set-encoder. Accordingly, set-representations that elicit a score surpassing a certain threshold are considered to originate from a novel environment.

Following the approach in [9], we consider the score distribution and set a threshold to classify a specific percentage, denoted as $q$, of in-distribution samples as originating from a known environment. To establish this threshold, we consider the $q$-th percentile of scores obtained from the validation

set. We also compare our novel environment detector with the same score function computed from singleton features $g(\mathbf{X})$ alone (see Table 2 for a preview).

# 3 Related Work

## 3.1 Domain Generalization

Domain Generalization (DG) trains models to perform under distribution shifts without access to test environments [5, 6]. In contrast, Domain Adaptation (DA) assumes test samples are available during training [14]. Both exploit multiple source domains, but DG is strictly test-agnostic. Non-marginal DG approaches fall into three groups [15]: data manipulation [16, 17], robust representation learning [18, 19], and learning strategy modification [20, 21] (see 15, 6 for reviews).

Between DA and DG lie test-time adaptation (TTA) and marginal transfer learning. TTA adapts to unlabeled test samples, often via fine-tuning or domain metadata [22, 23]. Marginal transfer instead assumes access to the marginal feature distribution $\frac{1}{n} \sum_i \sigma(\mathbf{X}_i)$ [7], with $\sigma$ implemented via CNNs [24], kernel embeddings [7, 25], or patch embeddings [26]. While Blanchard et al. [7] analyze kernel embeddings theoretically, existing work leaves open conditions for effectiveness, failure detection, and context-aware model selection. A recent alternative replaces permutation-invariant embeddings with transformers that exploit sample order [27].

Marginal transfer parallels in-context learning: labeled samples define context in the latter [28, 29], while unlabeled samples do so in the former. Finally, balancing in-domain and out-of-domain performance remains a central challenge [11, 12, 30]. Methods like Zhang et al. [30] mitigate this trade-off when domain identity is known, whereas our goal is to infer it.

## 3.2 Learning Permutation-Invariant Representations

Analyzing set-structured data with neural networks has received much theoretical [31, 8, 32] and empirical [33–35] momentum in recent years. For instance, [35] build on the set transformer architecture [34] and augment the attentive encoder with the capability to learn dynamic templates for attention-based pooling. Differently, [36] proposes to learn set-specific representations, along with global "prototypes", using an optimal transport (OT) optimization criterion.

A set-embedding can also be understood as a learned proxy variable for the confounder $E$. Generic proxy variables for confounding variables have been explored in the context of estimating the causal effect from $\mathbf{X}$ to $\mathbf{Y}$ in [37, 38]. While their work focuses on eliminating the effect of the confounding variable $E$, our objective is to leverage it for prediction purposes. Furthermore, they require $\mathbf{X}$ causing $\mathbf{Y}$ which does not conform to all prediction tasks. We do not require that $\mathbf{X}$ causes $\mathbf{Y}$ in our theoretical analysis and therefore include more scenarios (e.g., when $\mathbf{Y}$ is causing $\mathbf{X}$).

## 3.3 OOD Detection and Selective Classification

Detecting unusual inputs that deviate from the examples in the training set has been a long-standing problem of conceptual complexity in machine and statistical learning [39–43]. Flagging OOD instances involves identifying uncommon data points that might compromise the reliability of machine learning systems [40]. OOD detection is closely related to *inference with a reject option* (also termed selective classification) [44, 45], which allows classifiers to refrain from predicting ambiguous or novel conditions [46]. The reject option has been extensively studied in statistical and machine learning [47–50], with early work dating back to the 1950s [51, 52, 47].

More recently, [9] explored selective classification in DG settings. They investigated various *post-hoc scores* to define a "competence region" in feature space where a classifier is deemed competent. In this work, we consider a post-hoc score based on the $k$-nearest neighbours to the training set in feature space similar to [53], which applies to both classification and regression settings. Unlike the approach taken in [9], where the focus lies on features of individual instances, we consider the set summary provided by the set-encoder. Thus, we can identify novel environments even when singleton inputs lack sufficient information.

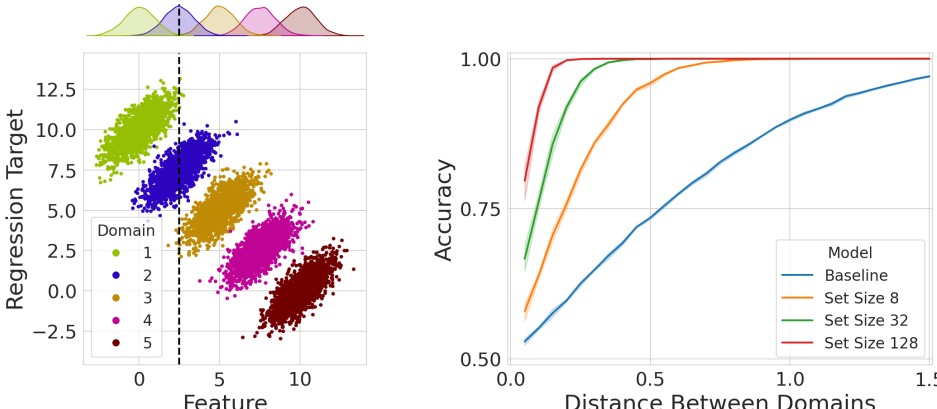

Figure 2: **Experiment 1**. *Left*: Toy dataset that conforms to our theoretical criteria. Without environmental information, the marked input at $x = 2.5$ could belong to either one of the domains numbered 1, 2, or 3 as indicated by the marginal distributions shown on top. *Right*: Comparison of environment classification accuracy for a baseline model versus a mean-pooled set-encoder using different set sizes. Distances between environments refer to the distance between the means of the environments. Smaller distances produce stronger overlap of the marginals. A detailed description can be found in Appendix E.1.

## 4 Experiments

In the following, we explore various aspects of context-aware models. First, we show on two datasets that a context-aware model achieves improved performance in ID and OOD settings compared to a baseline model when the necessary conditions of a source component shift are met. Second, we show how novel environments can be detected to select between the most predictive (in the ID setting) and the most robust (in the OOD setting) model. We also show that novel environment detection can be utilized to avoid failure cases. Third, we demonstrate that the necessary criteria (see Section 2.3) can be validated empirically, identifying cases where no benefits of the method can be expected. Experimental details can be found in the **Appendix** and the source code is available at [2].

### 4.1 Evaluation Approach

To approximate Criterion 2.1, Criterion 2.2, and Criterion 2.3, we train five models (Table 1). Our *context-aware model* $f^{\mathbf{Y}|\mathbf{X},\mathbf{S}^{(n)}}$ leverages context sets, while the *baseline* $f^{\mathbf{Y}|\mathbf{X}}$ ignores them. Their relative improvement is

$$\mathcal{R}_{\mathrm{I}} = \frac{\mathcal{M}(f^{\mathbf{Y}|\mathbf{X},\mathbf{S}^{(n)}}) - \mathcal{M}(f^{\mathbf{Y}|\mathbf{X}})}{\mathcal{M}(f^{\mathbf{Y}|\mathbf{X}})}, \tag{2}$$

where $\mathcal{M}(\cdot)$ is a test performance metric (negative L2-loss for regression). $\mathcal{R}_{\mathrm{I}} > 0$ indicates that Criterion 2.1 holds. For Criterion 2.2, we compare the contextual environment predictor $f^{E|\mathbf{X},\mathbf{S}^{(n)}}$ with its baseline $f^{E|\mathbf{X}}$:

$$\mathcal{R}_{\mathrm{II}} = \frac{\mathcal{M}(f^{E|\mathbf{X},\mathbf{S}^{(n)}}) - \mathcal{M}(f^{E|\mathbf{X}})}{\mathcal{M}(f^{E|\mathbf{X}})}. \tag{3}$$

We set $n$ such that $f^{E|\mathbf{X},\mathbf{S}^{(n)}}$ achieves nearly perfect ID accuracy; $\mathcal{R}_{\mathrm{II}} > 0$ supports Criterion 2.2. Finally, to test Criterion 2.3, we introduce the environment-oracle model $f^{\mathbf{Y}|\mathbf{X},E}$ and compute

$$\mathcal{R}_{\mathrm{III}} = \frac{\mathcal{M}(f^{\mathbf{Y}|\mathbf{X},E}) - \mathcal{M}(f^{\mathbf{Y}|\mathbf{X}})}{\mathcal{M}(f^{\mathbf{Y}|\mathbf{X}})}. \tag{4}$$

These relative improvement metrics serve as proxies for the theoretical criteria: when $\mathcal{M}$ is cross-entropy under optimal models, they align with mutual information measures. However, if $\mathcal{M}$ accuracy, then this proxy is not isomorphic to the criterion it approximates.

---

[2]`https://github.com/LarsKue/context-aware-domain-generalization`

| Model | Symbol | Description | Purpose |
|---|---|---|---|
| Context-aware (ours) | $f^{\mathbf{Y}|\mathbf{X},\mathbf{S}^{(n)}}$ | Predicts $\mathbf{Y}$ from $\mathbf{X}$ and $\mathbf{S}^{(n)}$ | Test Criterion 2.1 |
| Baseline | $f^{\mathbf{Y}|\mathbf{X}}$ | Predicts $\mathbf{Y}$ from $\mathbf{X}$ | Reference |
| Environment-oracle | $f^{\mathbf{Y}|\mathbf{X},E}$ | Predicts $\mathbf{Y}$ from $\mathbf{X}$ and $E$ | Test Criterion 2.3 |
| Contextual env. | $f^{E|\mathbf{X},\mathbf{S}^{(n)}}$ | Predicts $E$ from $\mathbf{X}$ and $\mathbf{S}^{(n)}$ | Test Criterion 2.2 |
| Baseline env. | $f^{E|\mathbf{X}}$ | Predicts $E$ from $\mathbf{X}$ | Reference for Criterion 2.2 |

Table 1: Five models used to evaluate our approach and verify the theoretical criteria.

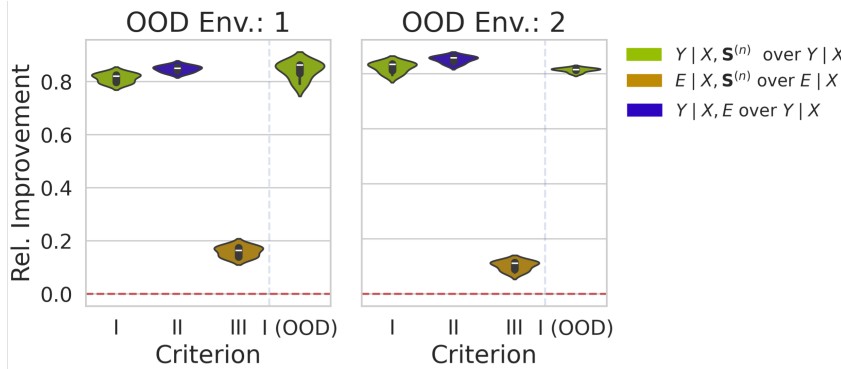

Figure 3: **Experiment 1**. Relative improvement of marginal transfer learning (shown in I) versus a baseline model (0 means no improvement is achieved) on a toy example. We also show I (OOD) on OOD data. II depicts the relative improvement of the environment-oracle model compared to the baseline model. III demonstrates the relative improvement in predicting the environment when using contextual information compared to the absence of it. Sampling variation arises from using different seeds to partition the ID data into training, test, and validation sets.

## 4.2 Experiment 1: Toy Example

**Setup**  To set the stage, we consider a dataset shown in Figure 2. The dataset includes data from five different environments, defined by distinct Gaussian distributions. Each Gaussian deviates due to its location (i.e. mean vector). The dataset exemplifies Simpson's paradox, wherein fitting without accounting for environmental factors would yield a negatively sloped line. This trend reverses to multiple positively sloped lines when considering environmental factors (see Figure 11).

Importantly, the dataset meets our necessary criteria, since the environment cannot be inferred from a single input as indicated by the overlap of the marginal distributions in Figure 2. The mathematical details underlying this dataset are described in Appendix E.1.

**Results**  As a first check of Criterion 2.2, we evaluate whether a set input provides additional information about the environment compared to a singleton input. Figure 2 illustrates that additional set input improves the ability to distinguish between environments significantly and the more samples we include, the better the distinction. As expected, a decrease in the distance between environment marginal means necessitates more samples to differentiate between environments.

Next, we assess the predictive capabilities of the context-aware approach across all possible scenarios of "leave-one-environment-out". This involves training on all environments except one and treating the excluded environment as a novel OOD scenario. Here, we consider linear models to ensure an optimal inductive bias for the problem. We can see that Criterion 2.1, Criterion 2.2 and Criterion 2.3 are satisfied in Figure 3. Providing contextual information in the form of a set input increases the performance significantly compared to a baseline model in the ID as well as in the OOD setting (see I and I (OOD) in Figure 3). We also observe a slightly higher relative improvement when the environment label is directly provided (see II) compared to using the output of the set-encoder (see I). This aligns with our expectations, as the set input does not offer more information about the target value than the environment label itself. Note that for metric III, we achieve less relative improvement since we consider the accuracy and not the L2-Loss.

|  | Accuracy [%] ↑ | |
| | ID | OOD |
|---|---|---|
| Baseline | $\mathbf{84.6} \pm 0.3$ | $10.2 \pm 0.3$ |
| Invariant | $72.8 \pm 0.9$ | $\mathbf{73.1} \pm 0.2$ |
| Selection (Ours) | $\mathbf{84.1} \pm 0.3$ | $\mathbf{73.1} \pm 0.2$ |
| Selection (Baseline) | $\mathbf{84.0} \pm 0.3$ | $14.0 \pm 0.4$ |
| Bayes Optimal | 85.0 | 75.0 |

Table 2: **Experiment 2**. Accuracy across model types and domain settings. Our context-aware model yields improved OOD detection compared to the baseline, allowing model selection at inference time. See Appendix K for more details.

In Appendix E all scenarios where one environment is left out for testing can be found. Additionally, we present there similar results for non-linear models and also demonstrate that the specific choice of permutation-invariant network does not significantly impact the prediction of the environment label. Furthermore, in Appendix B, we conduct an additional experiment resembling **Experiment 1**, but with high-dimensional inputs and achieve similar results.

### 4.3 Experiment 2: Colored MNIST

**Setup** The ColoredMNIST dataset [54] is an extension of the standard MNIST dataset, wherein the number of classes is reduced to two classes (digits $< 5$ and $\geq 5$). Furthermore, label noise is deliberately added, such that only in 75% of all cases, the label can be correctly predicted from the shape. To make things more challenging, the image background can take two colors that are also associated with the image label. In the first environment, the association is 90% and in the second one 80%. Therefore, a baseline model would tend to utilize the background for prediction instead of the actual shape. However, in a third environment, the associations are reversed, so that a model based on the background color would achieve only 10% accuracy (i.e., worse than random).

This dataset implies a trade-off between predictive performance in ID domains versus robustness in OOD domains, as discussed in [54, 30]. For instance, an invariant model that relies solely on an object's shape would be robust to domain shift at the cost of lower accuracy in the first two environments (75% vs. 80% or 90%). In contrast, a baseline model would achieve greater accuracy in the first domains (80% and 90%), but would fail dramatically in the third domain (only 10%).

**Results** Here, we assume the invariant model to be given, but it could also be obtained by invariant learning, e.g. Invariant Risk Minimization [54]. With our novel environment detection approach (see Section 2.5) we can get the best of both worlds, circumventing the inherent trade-off. When identifying the ID setting, we utilize the baseline model that achieves the highest predictiveness within the observed environments. In case we detect the OOD setting, we employ the invariant model. We compare this kind of model selection due to the features $h_\psi(\mathbf{S}^{(n)})$ inherent to our model versus the features extracted by the baseline model.

The results can be found in Table 2. By utilizing model selection based on the set-summary $h_\psi(\mathbf{S}^{(n)})$, we nearly recover the ID accuracy while maintaining identical performance to the invariant model on OOD data. Evidently, the novel environment detection only works with set summaries. A feature extracted from a single sample does not provide enough information to reliably detect distribution shifts, leading to difficulties in effectively selecting between baseline and invariant model, as demonstrated in Table 2. Details on this experiment can be found in Appendix G.

### 4.4 Experiment 3: Violated Criteria

**Setup** To demonstrate the effects of criterion violation, we consider the PACS dataset [55], as well as the OfficeHome dataset [56], each with the Art environment chosen as the novel (OOD) domain.

**Results** As expected, when the criteria are not met, context-aware models cannot achieve a benefit over the baseline (see Figure 4). Validating the criteria empirically, we find that Criterion 2.2 is not satisfied for PACS, as a single sample is sufficient to infer the source domain with near-perfect accuracy. Furthermore, Criterion 2.3 is not satisfied, as Figure 4a depicts.

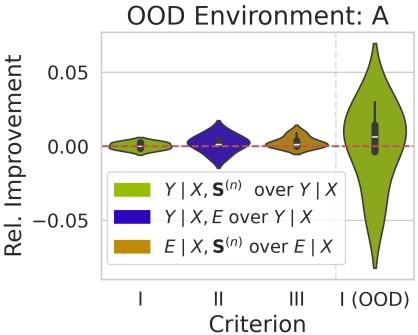 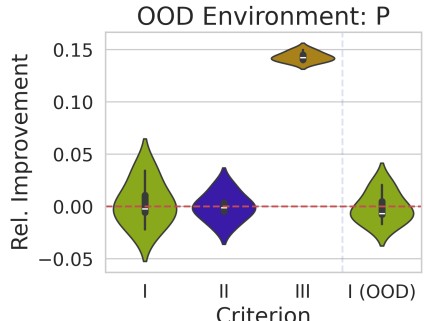

(a) Environment *Art* in PACS dataset. The environment is almost completely inferable from one input sample (Criterion 2.2 not satisfied). Conclusively the context-aware approach does not yield benefits.

(b) Environment *Product* in OfficeHome dataset. Although the environment is not inferable from one input sample (Criterion 2.2), the environment information does not yield benefits (Criterion 2.3).

Figure 4: **Experiment 3.** Tell-tale examples where at least one of the necessary criteria is not satisfied and the context-aware approach cannot possibly yield benefits.

| Winter | MSE $\downarrow$ | | AUROC |
|---|---|---|---|
| | ID | OOD | [%] $\uparrow$ |
| Baseline | $2.21 \pm 0.11$ | $6.08 \pm 0.13$ | $58.2 \pm 0.7$ |
| Ours | $\mathbf{2.09} \pm 0.12$ | $\mathbf{5.7} \pm 0.4$ | $\mathbf{100.0} \pm 0.0$ |

Table 3: **Experiment 4**. Inference performance (MSE) and novel environment detection (AUROC) comparison between our context-aware model and the baseline for the winter domain in the Bike-Sharing dataset. See Appendix K for more details.

On the OfficeHome dataset we find that Criterion 2.2 is not satisfied, while Criterion 2.3 is. Results are depicted in Figure 4b. We observe that the set input offers benefits for predicting the source environment corresponding to Criterion 2.3. However, even when providing the target classifier with the environment label, we do not achieve a benefit, suggesting that Criterion 2.2 is not satisfied. For experimental details, see Appendix H.

### 4.5 Experiment 4: Failure Case Detection

**Setup**  Besides unfulfilled criteria, another reason why a context-aware approach might fail to reap benefits is when the distribution shift requires extrapolation. This might be unattainable by the model, making the inclusion of a reject option beneficial. Using the BikeSharing dataset [57], we demonstrate that in cases where different seasons like summer or winter represent distinct environments, extrapolation might be necessary. We consider the task of predicting the number of bikes rented across the day based on weather data. Here we explore the scenario where we train on all seasons except winter. Details about the dataset, pre-processing steps, and other testing scenarios can be found in Appendix J.

**Results**  In Table 3 we demonstrate that the context-aware approach is slightly superior compared to the baseline model in the ID settings. However, both the baseline and the context-aware approach experience performance degradation in the novel winter environment. To detect the novel environment and, consequentially, potential failure cases, we compute the score as suggested in Section 2.5 and evaluate how well it distinguishes between ID versus OOD environments. We designate an independent ID test set and use the environment excluded during training (here winter) as the OOD set for evaluation. The area under the ROC-curve (AUROC) in Table 3 demonstrates that the score based on the permutation-invariant embedding allows for perfect detection of the novel environment, whereas the standard approach fails as expected.

# 5 Conclusions

In this work, we aimed to advance the theoretical understanding of marginal transfer learning in domain generalization. Accordingly, we formalized criteria that are necessary for context-aware models to yield benefits and are also verifiable in practice. Moreover, we pinpointed the source component shift as a scenario where context-aware models can offer advantages, enabling the identification of favorable settings and the identification of potential failure cases. The latter allows us to perform real-time model selection between the best performing model on ID data and the most robust (i.e., domain-invariant) model on OOD data. Future research should investigate generalization bounds, the learner's behavior in finite-data regimes, and the generalization behavior of the learner as the number of training domains increases (i.e., the domain efficieincy).

## Acknowledgments and Disclosure of Funding

This work is partially funded by the Deutsche Forschungsgemeinschaft (DFG, German Research Foundation) Project 528702768. JM and UK were supported by Informatics for Life funded by the Klaus Tschira Foundation. We thank Paul Christian Bürkner, Florian Fallenbüchel, Felix Draxler, and Armand Rousselot for their support and fruitful discussions.

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

# A  Pseudocode

---

**Algorithm 1:** Optimizing Equation 1 for context-aware domain generalization.

---

**Data:** Samples from the joint distribution $p(\mathbf{X}, \mathbf{Y}, E)$
**Input :** Composite model parameters $\boldsymbol{\theta}$, set size $n$, batch size $m$, loss-function $c$, number of iterations $k$, learning rate schedule $\alpha(k)$

1   **for** $i = 1, \ldots, k$ **do**
2     Sample mini-batch $\mathcal{B} = \{(\mathbf{x}_1, \boldsymbol{y}_1, \mathrm{env}_1), \ldots, (\mathbf{x}_m, \boldsymbol{y}_m, \mathrm{env}_m)\}$ from $p(\mathbf{X}, \mathbf{Y}, E)$
3     **for** $j = 1, \ldots, m$ **do**
4       Sample set $\mathbf{s}_j^{(n)} = \{\mathbf{x}_1, \ldots \mathbf{x}_n\}$ from $p(\mathbf{X} \mid E = \mathrm{env}_j)$
5       Replace $\mathrm{env}_j$ with $\mathbf{s}_j^{(n)}$ in $\mathcal{B}$
6     **end**
7     Update $\boldsymbol{\theta}$ using adaptive mini-batch gradient descent (or any variant):

$$\boldsymbol{\theta}_k \leftarrow \boldsymbol{\theta}_{k-1} - \alpha(k) \nabla_{\boldsymbol{\theta}} \left( \sum_{j=1}^{m} c \left( f_{\boldsymbol{\theta}}(\mathbf{x}_j, \mathbf{s}_j^{(n)}), \boldsymbol{y}_j \right) \right)$$

8   **end**
**Output :** Trained context-aware model $f_{\boldsymbol{\theta}}$

---

# B  Additional Experiment: ProDAS

**Setup**   We utilize the ProDAS library [58] to generate high-dimensional image data that meets our dataset requirements. The dataset comprises objects of shape square and circle, exhibiting variations in their texture, background color, rotation, and size. Additionally, the background varies in color and texture, resulting in a complex scenario. For examples see Figure 10. We consider the task of predicting the object size. Difficulties arise due to the presence of distinct environments with varying characteristics. Specifically, depending on the environment, a constant is added to the observed object size to get the actual target variable that we aim to predict:

$$Y_{\mathrm{gt}} = Y_{\mathrm{observed}} + j \cdot \mathrm{const}_1 \tag{5}$$

Here, $j \in \{1, 2, 3, 4\}$ denotes the environment, while $Y_{\mathrm{gt}}$ represents the ground truth (or factual) size, obtained as a sum of the observed size $Y_{\mathrm{observed}}$ (relative to the image frame) and a constant depending on $j$.

The background color follows a normal distribution $\mathcal{N}(\boldsymbol{\mu}_j; \boldsymbol{\Sigma})$ where the mean depends on the environment in the following way: $\boldsymbol{\mu}_j = \boldsymbol{\mu}_0 + j \cdot \mathrm{const}_2$. Here we assign a small value to $\mathrm{const}_2$ to enforce the background distributions to overlap between different environments. Specifically, this construction implies that the relation between input $\mathbf{X}$ and target $Y$ differs across environments. This corresponds to Criterion 2.3. Notably, inferring the originating environment from a single sample is unattainable due to overlapping background distributions (corresponding to Criterion 2.2). Samples of different environments are shown in Appendix F. This example could be inspired by microscopy data where different microscopes correspond to distinct environments, each exhibiting its own characteristics. During training, we assume to have access to the ground truth value $Y_{\mathrm{gt}}$.

**Results**   In line with the results from the previous toy example, we can demonstrate a strong relative improvement in the ProDAS dataset, as depicted in Figure 5. All formal criteria are satisfied and a very significant improvement is achieved, both in the ID and the OOD setting, by considering the contextual information from the environment. Additional details for this experiment can be found in Appendix F.

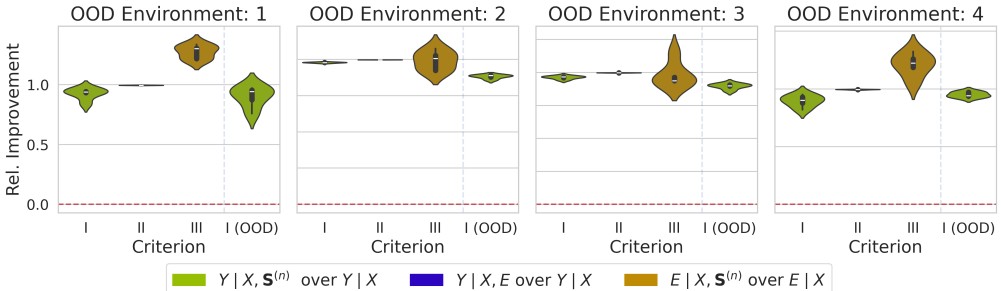

Figure 5: **Experiment 2**: Relative improvement of set-encoder (shown in I) approach versus baseline model (0 means, no improvement is achieved) on ProDAS dataset. We also show I (OOD) on OOD data. II depicts the relative improvement of the environment-oracle model compared to the baseline model. III demonstrates the relative improvement in predicting the environment when using contextual information compared to the absence of it. Variations arise from using different seeds to partition the ID data into training, test and validation set.

## C  Theory

### C.1  Generalization of Theorem 2.1 to Noisy Environments

**Theorem C.1.** *In addition to Theorem 2.1, the following holds:*

*(d) Assume that there exists a function $g$ and a noise variable $Z$ that elicits the relation $E = g(\mathbf{S}^{(n)}) + Z$ and satisfies $\mathbf{S}^{(n)} \perp Z \mid \mathbf{X}$ as well as $\mathbf{S}^{(n)} \perp Z \mid \mathbf{X}, Y$. Furthermore, assume that $\mathbf{Y} \not\perp E \mid \mathbf{X}$ and $I(\mathbf{Y}; E \mid \mathbf{X}) > I(Z; \mathbf{Y} \mid \mathbf{X})$. Then, we achieve $\mathbf{Y} \not\perp \mathbf{S}^{(n)} \mid \mathbf{X}$, recovering Criterion 2.1.*

The proof can be found in Appendix C.3.

### C.2  Insufficiency of Criteria 2 and 3 for Criterion 1

Criterion 2.2 and Criterion 2.3 are not sufficient to imply Criterion 2.1. This can be seen in an example with three environments $j \in \{1, 2, 3\}$. Assume the first two have completely identical input distributions. We presume that both input distributions adhere to a uniform distribution $\mathcal{U}[a, b]$. Furthermore, we assume that the third input distribution also follows a uniform distribution that is slightly shifted, i.e. $\mathcal{U}[a + \frac{a+b}{2}, b + \frac{a+b}{2}]$. Due to the overlap between the third and the first two environments, a set input provides additional information about $E$ compared to a single sample $X$, verifying Criterion 2.2.

Regarding the mechanism relating inputs to outputs, we assume that on $[a, \frac{a+b}{2}]$ the relation between input $X$ and output $Y$ differs, e.g., two constant functions with distinct values. We further assume that on $(\frac{a+b}{2}, b + \frac{a+b}{2}]$ the relation between input $X$ and output $Y$ does not vary with the environment, e.g., is constant. This aligns with Criterion 2.3: if we know the environment, we can improve the prediction, specifically on $[a, \frac{a+b}{2}]$.

However, Criterion 2.1 is not satisfiable. The set input allows us to distinguish environment 3 (i.e. the one with support $\mathcal{U}[a + \frac{a+b}{2}, b + \frac{a+b}{2}]$) from the other ones. Yet, we cannot distinguish between environment 1 and environment 2. Since the relation between $X$ and output $Y$ differs only in the supports of environment 1 and environment 2 (specifically, it differs in $\mathcal{U}[a, \frac{a+b}{2}]$), the set input cannot provide additional information about the output $Y$ compared to the single input $X$, i.e. it holds $Y \perp \mathbf{S}^{(n)} \mid X$.

It is also worth noting that Criterion 2.3 might be achievable while Criterion 2.2 is unattainable and vice versa. For instance, when we can infer the originating environment from one sample (Criterion 2.2 is not attainable), the relation between $\mathbf{X}$ and $\mathbf{Y}$ might still vary with the environment (Criterion 2.3 is achievable).

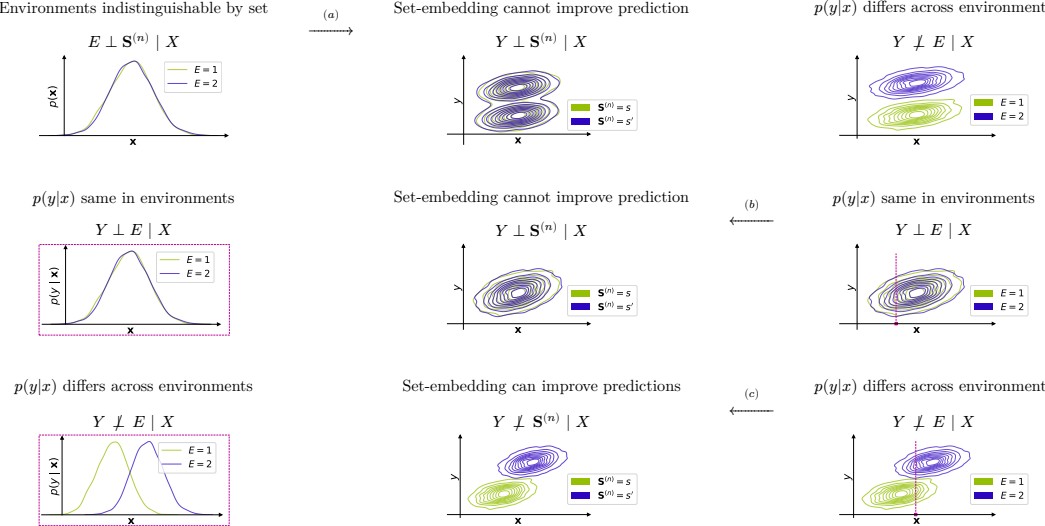

Figure 6: Illustration of Theorem 2.1. The first row depicts (a), the second row (b) and the third row (c). The pink framed plots show the conditional distributions along the pink marker as shown on the right.

## C.3 Illustration and Proof of Theorem 2.1

In the following, we give proofs of Theorem 2.1 (a) - (d).

*Proof.* For the upcoming proofs, we extensively employ the chain rule of mutual information:

$$I(\mathbf{Y}; Z, \mathbf{X}) = I(\mathbf{Y}; Z \mid \mathbf{X}) + I(\mathbf{Y}; \mathbf{X}) \tag{6}$$

Additionally, we have the inequalities $I(\mathbf{Y}; \mathbf{S}^{(n)} \mid \mathbf{X}) \leq I(\mathbf{Y}; E \mid \mathbf{X})$ and $I(\mathbf{S}^{(n)}; \mathbf{Y} \mid \mathbf{X}) \leq I(E; \mathbf{Y} \mid \mathbf{X})$ that follow from the data processing inequality and how $\mathbf{S}^{(n)}$ relates to the other variables (see Figure 1).

For (b): We easily achieve

$$I(Y; \mathbf{S}^{(n)}, \mathbf{X}) = I(\mathbf{Y}; \mathbf{S}^{(n)} \mid \mathbf{X}) + I(\mathbf{Y}; \mathbf{X}) \tag{7}$$
$$\leq I(\mathbf{Y}; E \mid \mathbf{X}) + I(\mathbf{Y}; \mathbf{X}) \tag{8}$$
$$= I(\mathbf{Y}; \mathbf{X}) \tag{9}$$

Therefore, we have

$$0 \leq I(\mathbf{Y}; \mathbf{S}^{(n)} \mid \mathbf{X}) = I(\mathbf{Y}; \mathbf{S}^{(n)}, \mathbf{X}) - I(\mathbf{X}; \mathbf{Y}) \leq 0 \tag{10}$$

which proves (b).

For (a): We can write

$$I(\mathbf{S}^{(n)}; \mathbf{Y}, \mathbf{X}) = I(\mathbf{S}^{(n)}; \mathbf{Y} \mid \mathbf{X}) + I(\mathbf{S}^{(n)}; \mathbf{X}) \tag{11}$$
$$\leq I(\mathbf{S}^{(n)}; E \mid \mathbf{X}) + I(\mathbf{S}^{(n)}; \mathbf{X}) \tag{12}$$
$$= I(\mathbf{S}^{(n)}; \mathbf{X}) \tag{13}$$

and therefore

$$0 \leq I(\mathbf{Y}; \mathbf{S}^{(n)} \mid \mathbf{X}) = I(\mathbf{S}^{(n)}; \mathbf{Y}, \mathbf{X}) - I(\mathbf{X}; \mathbf{S}^{(n)}) \leq 0 \tag{14}$$

and conclusively $\mathbf{Y} \perp \mathbf{S}^{(n)} \mid \mathbf{X}$.

For (c) is easily seen that $0 < I(\mathbf{Y}; E \mid \mathbf{X}) = I(\mathbf{Y}; g(\mathbf{S}^{(n)}) \mid \mathbf{X}) \leq I(\mathbf{Y}; \mathbf{S}^{(n)} \mid \mathbf{X})$ and therefore (c) holds true.

For (d), we also employ the entropy $h(\mathbf{X})$ as well as the conditional entropy $h(\mathbf{X} \mid \mathbf{Y})$. We first establish that $I(A + B; C) \leq I(A; C) + I(B; C)$ for any RVs $A, B, C$ with $A \perp B$ and $A \perp B \mid C$:

$$I(A + B; C) = h(A + B) - h(A + B \mid C)$$

$$\overset{(\star)}{=} (h(A) + h(B) - h(A \mid A + B)) - (h(A \mid C) + h(B \mid C) - h(A \mid A + B, C))$$

$$= I(A; C) + I(B; C) - h(A \mid A + B) + h(A \mid A + B, C)$$

$$\overset{(\star\star)}{\leq} I(A; C) + I(B; C) \tag{15}$$

$(\star)$ follows with the chain rule for entropy

$$h(A, A + B) = h(A) + h(A + B \mid A) \tag{16}$$

$$= h(A) + h(B \mid A) \overset{A \perp B}{=} h(A) + h(B) \tag{17}$$

$$= h(A + B) + h(A \mid A + B) \tag{18}$$

which implies $h(A + B) = h(A) + h(B) - h(A \mid A + B)$ and equally when conditioning on $C$.

$(\star\star)$ follows since $h(A \mid A + B, C) \leq h(A \mid A + B)$.

Equation 15 can be extended to the conditional mutual information if $A \perp B \mid D$ and $A \perp B \mid D, C$:

$$I(A + B; C \mid D) \leq I(A; C \mid D) + I(B; C \mid D) \tag{19}$$

Since $\mathbf{S}^{(n)} \perp Z \mid \mathbf{X}$ and $\mathbf{S}^{(n)} \perp Z \mid \mathbf{X}, Y$, we achieve

$$0 < I(\mathbf{Y}; E \mid \mathbf{X}) = I(Y; g(\mathbf{S}^{(n)}) + Z \mid \mathbf{X}) \tag{20}$$

$$\leq I(\mathbf{Y}; g(\mathbf{S}^{(n)}) \mid \mathbf{X}) + I(\mathbf{Y}; Z \mid \mathbf{X}) \tag{21}$$

$$\leq I(\mathbf{Y}; \mathbf{S}^{(n)} \mid \mathbf{X}) + I(\mathbf{Y}; Z \mid \mathbf{X}) \tag{22}$$

and therefore

$$0 < I(\mathbf{Y}; E \mid \mathbf{X}) - I(\mathbf{Y}; Z \mid \mathbf{X}) \leq I(\mathbf{Y}; \mathbf{S}^{(n)} \mid \mathbf{X}) \tag{23}$$

which concludes the proof. $\qquad\square$

In the following, we discuss the assumptions in (c) and (d). In our experiments, we observed that in most datasets a relatively small sample size suffices to infer the environment label with approximately 100% accuracy (see Table 6). Therefore, the assumption that there exists a function $g(\mathbf{S}^{(n)}) = E$ seems justified if $n$ is sufficiently large. To generalize the assumption where the environment label is not fully inferable, we have to make assumptions. For one, we require $\mathbf{S}^{(n)} \mid Z \mid \mathbf{X}$. This can be interpreted as "increasing the set size does not improve the prediction of $E$". Also $\mathbf{S}^{(n)} \perp Z \mid \mathbf{X}, Y$ can be interpreted similarly: increasing the set size and considering the ground truth label/value does not enhance the predictability of $E$. Both assumptions should hold approximately if $n$ is large enough. With the assumption $I(\mathbf{Y}; E \mid \mathbf{X}) > I(Z; \perp \mathbf{Y} \mid \mathbf{X})$ we assume that the noise $Z$ is less predictive of $Y$ compared to $E$ if $\mathbf{X}$ is given. This can be roughly interpreted as the noise does not prove useful for predicting $Y$ from $\mathbf{X}$ compared to the ground truth environment label.

## D   Experiments: General Remarks

We define the relative improvements $\mathcal{R}_{\text{II}}$ and $\mathcal{R}_{\text{III}}$ as

$$\mathcal{R}_{\text{II}} = \frac{\mathcal{M}(f^{E|\mathbf{X}, \mathbf{S}^{(n)}}) - \mathcal{M}(f^{E|\mathbf{X}})}{\mathcal{M}(f^{E|\mathbf{X}})} \tag{24}$$

and

$$\mathcal{R}_{\text{III}} = \frac{\mathcal{M}(f^{\mathbf{Y}|\mathbf{X}, E}) - \mathcal{M}(f^{\mathbf{Y}|\mathbf{X}})}{\mathcal{M}(f^{\mathbf{Y}|\mathbf{X}})} \tag{25}$$

$\mathcal{R}_{\mathrm{II}}$ signifies the relative performance gain in predicting the environment when the set input is given compared to the solitude input. In contrast, $\mathcal{R}_{\mathrm{III}}$ denotes the relative performance improvement of the environment-oracle model compared to the baseline model.

Due to the large amount of settings, we did only little hyper-parameter optimization (we looked into batch size, learning rate, and network size). For a given dataset we optimized only on one scenario where an environment is left out during training. The found hyper-parameters were then applied to all other scenarios. To ensure that the baseline model is comparable to ours, we ensure that the inference network (and feature extractor) in Figure 1 have a comparable number of parameters as the baseline model. In all cases, the set-encoder is kept simple and its hyper-parameters are selected for optimal performance of the contextual environment predictor $f^{E|\mathbf{X},\mathbf{S}^{(n)}}$. For an overview, see Table 6. Throughout all experiments, we employ a mean-pooling operation.

We show the accuracies of classifying the environment of the contextual-environment model $f^{E|\mathbf{X},\mathbf{S}^{(n)}}$ and the baseline environment model $f^{E|\mathbf{X}}$ in Table 6. Here we only consider the datasets where we performed a full evaluation of all criteria.

### D.1  Computational complexity

We run all experiments using four Titan X GPUs, with 12GB VRAM each. On this hardware, each experiment requires between two and three days to run to completion. Our code base provides several utilities to reduce the overall memory footprint, allowing reproduction of our experiments on less powerful hardware.

## E  Experiment 1: Details

### E.1  Data Generation

Simpson's Paradox [3, 4] describes a statistical phenomenon wherein several groups of data exhibit a trend, which reverses when the groups are combined. There are several famous real-world examples of Simpson's Paradox, such as a study examining a gender bias in the admission process of UC Berkeley [59] or an evaluation of the efficacy of different treatments for kidney stones [60].

To replicate this, we create a dataset inspired by an illustration of Simpson's Paradox on Wikipedia [61]. The dataset consists of a mixture of 2D multivariate normal distributions, with the intent of using the first dimension as a feature, and the second as a regression target. Unless otherwise specified, we generate the data by taking an equal number of samples from each mixture component, defining the environment as a one-hot vector over the mixture components.

The mixture components are chosen to lie on a trend line that is opposite to the trend within each mixture. We achieve this by using a negative global trend and choosing the covariance matrix of each mixture as a scaled and rotated identity matrix with an opposite trend.

| Setting | Value | Controls |
|---|---|---|
| n_domains | 5 | number of mixture components |
| n_samples | 10000 | number of samples per mixture component |
| spacing | 2.0 | spacing between means of the mixture components |
| noise | 0.25 | overall noise level |
| noise_ratio | 6.0 | ratio of the primary to secondary noise axis |
| rotation_range | $(45.0, 45.0)$ | min (leftmost) and max (rightmost) mixture rotation angle |

Table 4: Default Settings for the Simpson's Paradox Dataset. Samples from the dataset constructed with these settings can be seen in Figure 2

The YouTube channel minutephysics also published a short descriptive video on this phenomenon in 2017 [62].

### E.2  Training Details

We consider five distinct settings, where in each setting, one domain is left out during training, and considered for evaluation as a novel environment. To gauge the uncertainty stemming from data

sampling, we also consider five dataset seeds for partitioning into training, validation, and test sets. For each dataset seed and model, we consider the results due to the best performance on the validation set.

We enforced that our approach and the baseline model have a similar amount of parameters for the feature extractor and final inference model. We conducted minimal hyperparameter tuning (focusing on parameters such as the learning rate schedule, batch size, and the number of parameters), and this was performed solely within one "leave-one-environment-out" setting. In total, we trained the five models outlined in Table 1 using five distinct dataset seeds. Consequently, a total of $5 \cdot 5 \cdot 5 = 125$ models were trained. In all cases, the set-encoder is kept simple and its hyper-parameters are selected for optimal performance of the contextual environment predictor $f^{E|\mathbf{X},\mathbf{S}^{(n)}}$. We choose the mean as the pooling operation.

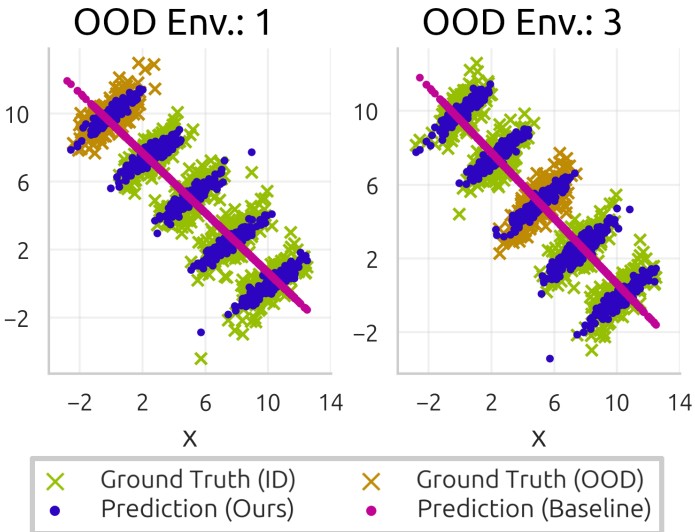

Figure 7: **Experiment 1**. Predictions performed on the toy dataset illustrated in Figure 2. We show predictions made by both our set-encoder approach and the vanilla model in the ID and OOD settings.

Now, we visualize the predictions of the baseline approach and our set-encoder approach in Figure 7 for one trained model. Our model captures and utilizes the characteristics of each environment for prediction. In contrast, the baseline approach struggles to discern between environments due to the significant overlap between environments, resulting in an inability to deal with environmental differences. Note that we obtained the best results by considering a class of linear models that aligns with the data-generating process. However, we observe that extrapolation performance drops when the considered models are overly complex and lack a strong inductive bias (see Appendix E.3).

### E.3   Non-Linear Models

In the experiments in Section 4.2, we considered linear models for our model and the baseline. In the following, we show results for the non-linear model class in Figure 8. We compare predictions of a baseline model and our model on all environments in Figure 9. We see that the extrapolation task fails in some cases as in environment 1. This is due to the mismatch of the considered model class and ground truth model.

## F   Additional Experiment: Details

Data samples from different environments are depicted in Figure 10. The process of how inputs relate to outputs is described in Appendix B.

During training, we employ a convolutional network to extract features $g(\mathbf{X})$. These features are passed to the inference network and the set-encoder. The feature extractor is then jointly trained with

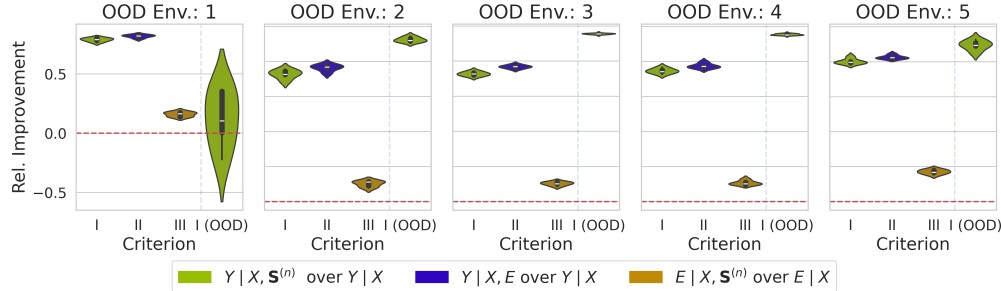

Figure 8: **Experiment 1.** Verification of criteria. In I we depict the relative improvement of our approach versus a baseline model. We also show I (OOD) on OOD data. In II we show the relative improvement of the oracle model compared to the baseline. In III we compare the relative improvement of the contextual environment model with respect to the baseline environment model.

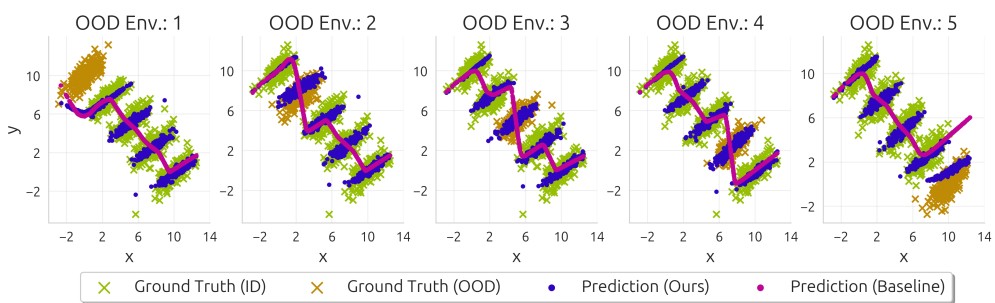

Figure 9: **Experiment 1.** Models are trained on all environments except the OOD environment. "Extrapolation", i.e. when environment 1 or 5 is OOD, is a particularly hard task in this setting. The set-based model shows slightly better extrapolation capabilities. Generally, our model exhibits adaptability to diverse environments, addressing a limitation present in the baseline model.

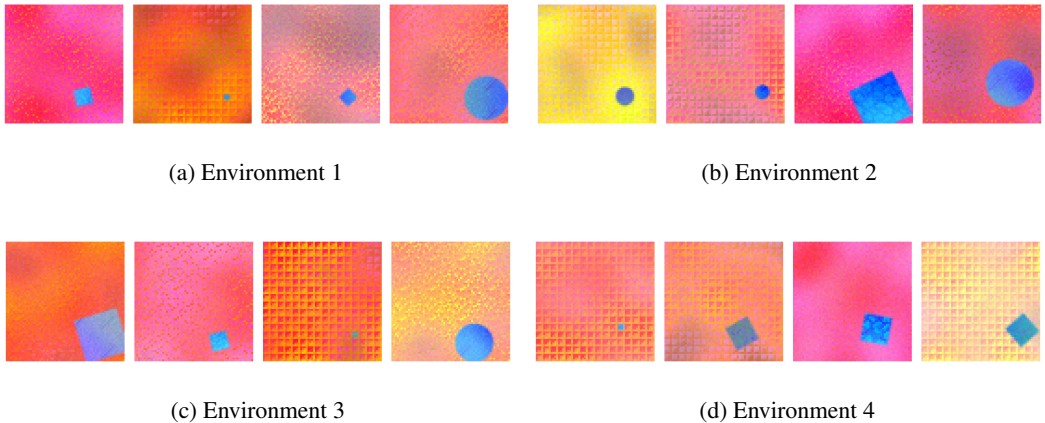

(a) Environment 1                (b) Environment 2

(c) Environment 3                (d) Environment 4

Figure 10: **Additional Experiment.** We generate four distinct domains synthetically. Notably, the background color within each domain follows a normal distribution. However, there are variations in the means across these domains Note that there is a huge overlap between the environments.

the inference network and set-encoder. We ensured that the feature extractor plus inference network and the baseline model have a comparable amount of parameters. The set-encoder is kept simple and its hyper-parameters are selected for optimal performance of the contextual environment predictor $f^{E|\mathbf{X},\mathbf{S}^{(n)}}$. As a pooling operation we choose the mean-pooling.

## G   Experiment 2: Details

To select between the baseline model and the invariant model, we are required to distinguish between ID and OOD data. Therefore, we follow the approach proposed in Section 2.5. We consider the $k$-nearest neighbors of the training set to compute the score $s_\psi$ where $k = 5$. Since we compare the scores elicited by features of the baseline model with the scores elicited by the features extracted by the set-encoder, we restricted both architectures to have the same feature dimension. To establish a threshold for distinguishing between ID and OOD samples, we designate samples with scores below the 95% quantile of the validation set as ID and those above as OOD (see Section 2.5 for details).

In total, we explore five dataset seeds to partition into training, validation, and test sets. To train an invariant model, we considered the same split in training, validation, and test set where the background color has no association with the label. Therefore the invariant model learns to ignore the background color and only utilize the shape for prediction. To learn effectively about the environment, we considered a large set input, namely 1024 samples in $\mathbf{S}^{(n)}$. We employed a simple set-encoder incorporating a mean pooling operation.

## H   Experiment 3 and 4: Details

For the BikeSharing dataset we consider a simple feed-forward neural network in all models. For the PACS as well as the OfficeHome dataset we consider features $g(\mathbf{X})$ that are kept fixed and not optimized. Here, we employ the Clip features proposed in [63]. The inference model, baseline model, and set-encoder are kept simple and employ only linear layers followed by ReLU activation functions. Given that Clip features considerably simplify the task, we performed a minimal hyper-parameter search and ensured that the inference model had a similar number of parameters as the baseline model. In all cases, the set-encoder is kept simple and its hyper-parameters are selected for optimal performance of the contextual environment predictor $f^{E|\mathbf{X},\mathbf{S}^{(n)}}$.

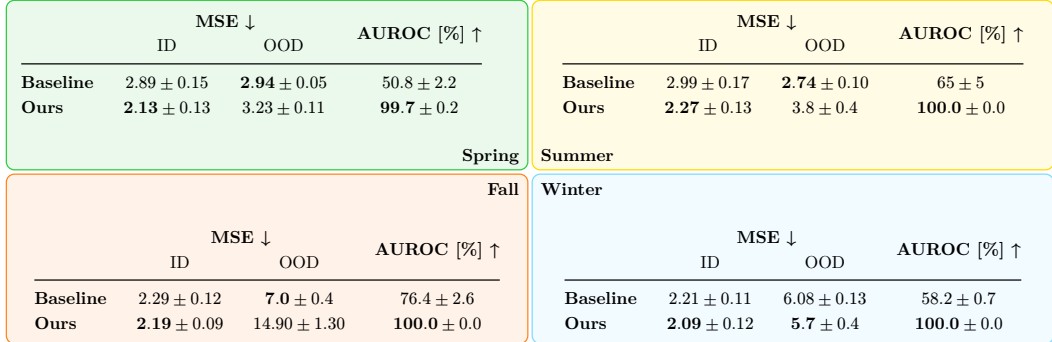

|  | MSE ↓ | | AUROC [%] ↑ |
|---|---|---|---|
|  | ID | OOD |  |
| Baseline | $2.89 \pm 0.15$ | $\mathbf{2.94 \pm 0.05}$ | $50.8 \pm 2.2$ |
| Ours | $\mathbf{2.13 \pm 0.13}$ | $3.23 \pm 0.11$ | $\mathbf{99.7 \pm 0.2}$ |

Spring

|  | MSE ↓ | | AUROC [%] ↑ |
|---|---|---|---|
|  | ID | OOD |  |
| Baseline | $2.99 \pm 0.17$ | $\mathbf{2.74 \pm 0.10}$ | $65 \pm 5$ |
| Ours | $\mathbf{2.27 \pm 0.13}$ | $3.8 \pm 0.4$ | $\mathbf{100.0 \pm 0.0}$ |

Summer

Fall

Winter

|  | MSE ↓ | | AUROC [%] ↑ |
|---|---|---|---|
|  | ID | OOD |  |
| Baseline | $2.29 \pm 0.12$ | $\mathbf{7.0 \pm 0.4}$ | $76.4 \pm 2.6$ |
| Ours | $\mathbf{2.19 \pm 0.09}$ | $14.90 \pm 1.30$ | $\mathbf{100.0 \pm 0.0}$ |

|  | MSE ↓ | | AUROC [%] ↑ |
|---|---|---|---|
|  | ID | OOD |  |
| Baseline | $2.21 \pm 0.11$ | $\mathbf{6.08 \pm 0.13}$ | $58.2 \pm 0.7$ |
| Ours | $\mathbf{2.09 \pm 0.12}$ | $5.7 \pm 0.4$ | $\mathbf{100.0 \pm 0.0}$ |

Table 5: **Experiment 4.** Performance comparison between our model and the baseline, broken down by target domain. We compare their performance in the ID and OOD setting (MSE), as well as their capability to detect a novel environment (AUROC). Both models experience a performance drop in the OOD setting, but our model can detect with strong certainty when this is the case. See Appendix K for more details.

In all cases, the set-encoder is kept simple and its hyper-parameters are selected for optimal performance of the contextual environment predictor $f^{E|\mathbf{X},\mathbf{S}^{(n)}}$.

| Dataset / Set Size | Simpson / 32 | | | | |
|---|---|---|---|---|---|
| Domain | 1 | 2 | 3 | 4 | 5 |
| $f^{E\mid\mathbf{X}}$ | $86.3 \pm 1.3$ | $90.8 \pm 1.3$ | $90.7 \pm 0.8$ | $90.4 \pm 0.9$ | $85.5 \pm 0.8$ |
| $f^{E\mid\mathbf{X},\mathbf{S}^{(n)}}$ | $\mathbf{100.0 \pm 0.0}$ | $\mathbf{100.0 \pm 0.0}$ | $\mathbf{100.0 \pm 0.0}$ | $\mathbf{100.0 \pm 0.0}$ | $\mathbf{100.0 \pm 0.0}$ |

| Dataset / Set Size | ProDAS / 128 | | | | OfficeHome / 4 | PACS / 4 |
|---|---|---|---|---|---|---|
| Domain | 1 | 2 | 3 | 4 | Product | Art |
| $f^{E\mid\mathbf{X}}$ | $43.8 \pm 1.1$ | $50.0 \pm 1.3$ | $49.9 \pm 2.3$ | $44.4 \pm 1.0$ | $86.16 \pm 0.33$ | $\mathbf{99.72 \pm 0.33}$ |
| $f^{E\mid\mathbf{X},\mathbf{S}^{(n)}}$ | $\mathbf{99.6 \pm 0.6}$ | $\mathbf{99.5 \pm 1.0}$ | $\mathbf{98.7 \pm 1.6}$ | $\mathbf{98.0 \pm 3.2}$ | $\mathbf{98.49 \pm 0.24}$ | $\mathbf{100.0 \pm 0.0}$ |

Table 6: Environment classification accuracy for different models and datasets, broken down by domain. As in Table 5, the uncertainty (mean and standard deviation) is computed over multiple seeds for dataset splits. In all cases, the set-based model outperforms the baseline.

# I    Comparison of Permutation-Invariant Architectures

As a pilot experiment, we estimate the contextual information contained in a set input by evaluating the binary classification accuracy of a set-based model compared to a baseline model with singleton sample input.

Importantly, we postulate that for stronger domain overlap, the contextual information contained within the single sample decreases significantly, while the contextual information within the set decreases only weakly, depending on the set size. Domains that do not overlap exactly will remain distinguishable, so long as the set size is large enough.

Therefore, we construct the toy dataset as described in Appendix E.1, but use the setting `n_domains = 2` and vary the distance between environments for each experiment.

We train each architecture on this dataset for 20 epochs, using 5 different seeds. We evaluate a total of 30 domain spacings, linearly distributed between $0.05$ and $1.5$ (both inclusive). Since we evaluate a baseline model, plus 3 set-based models at 3 different set sizes, this brings us to a total of $30 \cdot 20 \cdot 5 \cdot (1 + 3 \cdot 3) = 30000$ model epochs. We choose the batch size at 128 fixed.

Each architecture consists of a linear projection into a 64-dimensional feature space, followed by a fully connected network with 3 hidden layers, each containing 64 neurons and a ReLU [64] activation. For the set-based methods, this is followed by the respective pooling. We choose 8 heads for the attention-based model.

Finally, the output is linearly projected back into the 2-dimensional logit space, where the loss is computed via cross-entropy [65].

For methods that support a non-unit output set size, we choose the output set size as 4. The output set is mean-pooled prior to projection into the logit space.

# J    Bike Sharing Dataset

This dataset, taken from the UCI machine learning repository [57], consists of over 17000 hourly and daily counts of bike rentals between 2011 and 2012 within the Capital bike share system.

Each dataset entry contains information about the season, time, and weather at the time of rental. Casual renters are also distinguished from registered ones.

Similar to [66], we only consider the hourly rental data. We drop information about the concrete date and information about casual versus registered renters. We choose the season variable (spring, summer, fall, winter) as the environment and the bike rental count as the regression target. Since we deal with count data, we also apply square root transformation on the target similar to [66].

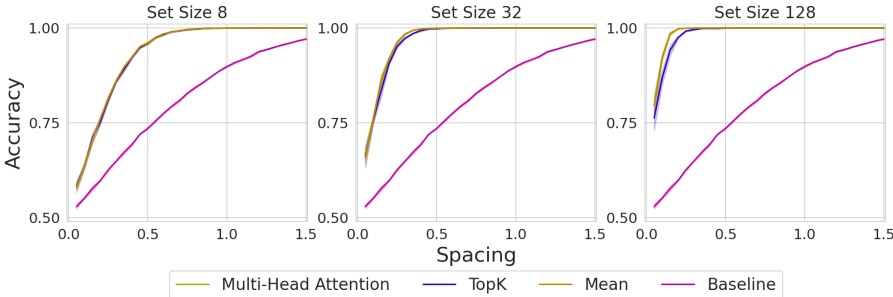

Figure 11: Comparison of different architectural choices for the permutation-invariant network in predicting the data's originating environment. We consider various distances between environments and different set sizes $n$. As anticipated, the plots illustrate that smaller environment distances make it more challenging to differentiate between them. Moreover, with a larger set size $n$, our ability to predict the environment label improves. Notably, the baseline model shows significantly poorer performance compared to the model utilizing contextual information in the form of a set input.

## K   Table Details

For tables 2, 3, and 5, we present the mean and standard deviation computed over 5 different training runs using separate seeds for partitioning the data into training, validation, and test sets.

We compute the AUROC by calculating a score for each sample as described in Section 2.5. The AUROC is then determined by calculating the AUC of the ROC curve, which is associated with the task of predicting the environment.

We highlight models within the 95% confidence interval of the best one for each respective category in bold.

## L   Potential Societal Impacts

This paper presents a foundational study, with societal impacts reliant mostly on the application of our methods. Nevertheless, we estimate that good-faith applications of our methods can have a positive societal impact. This manifests in improved performance results when our criteria are satisfied, as well as increased trustworthiness of these results due to the reliant detection of novel environments. This is particularly important for safety-critical applications, e.g., in medicine.

Negative societal impacts may also manifest in bad-faith applications, as the improved results may be misused. Furthermore, there is a risk that our methods may inadvertently perpetuate existing biases in data, particularly if environments are chosen in bad faith.

