# OpenReview forum: "Towards Context-Aware Domain Generalization: Understanding the Benefits and Limits of Marginal Transfer Learning"
_NeurIPS.cc/2025/Workshop/Reliable_ML — NeurIPS 2025 - Reliable ML Workshop_

### Official Review · Reviewer_YHjY · 2025-09-18
**Review: Limited Readability of Theoretical Contributions**

**Rating:** 7
**Confidence:** 1

**Review:**

Reviewer Expertise: I have limited expertise in the specific area of theoretical machine learning and found some of the more mathematical and theoretical aspects of this paper challenging to follow. My review is based on a high-level understanding of the core concepts and the empirical claims, rather than a full, in-depth technical validation.

Paper summary: This paper proposes a "context-aware" approach to the problem of domain generalization. The authors use a small set of data points from a new environment to create a context embedding, which is then used to improve model predictions. The work provides a theoretical analysis of the conditions under which this approach is beneficial and claims to offer a method to circumvent the well-known trade-off between predictive performance and robustness.

Strengths: This paper provides a valuable theoretical framework for understanding the benefits and limits of using context in domain generalization. The formalization of the problem and the establishment of verifiable necessary criteria for success are a significant contribution to the field. The paper also presents a method for detecting when a model is being used for out-of-distribution (OOD) extrapolation, which is a practical and useful insight.

Weaknesses/Limitations: The paper's highly theoretical nature makes it difficult for a reader without a strong background in the area to fully grasp the nuances of the analysis. While the core idea of using context is intuitive, the theoretical framework is challenging to follow without specialized knowledge, which may limit the paper's accessibility to a broader audience.

Suggestions for Authors: While the theoretical contribution is strong, it may benefit from additional, more intuitive examples to illustrate the key concepts and formalisms. A more in-depth discussion of the practical implications and trade-offs of implementing this approach in real-world applications would also enhance the paper's overall impact.